# A Robust Flow-Based System for the Spectrophotometric Determination of Cr(VI) in Recreational Waters

**DOI:** 10.3390/molecules27072073

**Published:** 2022-03-23

**Authors:** Tânia C. F. Ribas, Raquel B. R. Mesquita, Ana Machado, Joana L. A. Miranda, Graham Marshall, Adriano Bordalo, António O. S. S. Rangel

**Affiliations:** 1CBQF—Centro de Biotecnologia e Química Fina—Laboratório Associado, Escola Superior de Biotecnologia, Universidade Católica Portuguesa, Rua Diogo Botelho 1327, 4169-005 Porto, Portugal; tpedro@ucp.pt (T.C.F.R.); rmesquita@ucp.pt (R.B.R.M.); jmiranda@ucp.pt (J.L.A.M.); 2ICBAS—Instituto de Ciências Biomédicas Abel Salazar, Universidade do Porto, Rua Jorge de Viterbo Ferreira 228, 4050-313 Porto, Portugal; ammachado@icbas.up.pt (A.M.); bordalo@icbas.up.pt (A.B.); 3CIIMAR—UP, Interdisciplinary Centre of Marine and Environmental Research of the University of Porto, Novo Edifício do Terminal de Cruzeiros do Porto de Leixões, Avenida General Norton de Matos, S/N, 4450-208 Matosinhos, Portugal; 4Global FIA, Fox Island, WA 98333, USA; graham@globalfia.com

**Keywords:** sequential injection analysis, GlobalFIA, Cr(VI), DPC, dynamic water systems

## Abstract

A flow-based method for the spectrophotometric determination of chromium (VI) in recreational waters with different salinities was developed. Chromium can occur in the environment in different oxidation states with different related physiological properties. With regard to chromium, the speciation is particularly important, as the hexavalent chromium is considered to be carcinogenic. To achieve that purpose, the use of the diphenylcarbazide (DPC) selective colored reaction with the hexavalent chromium was the chosen strategy. The main objective was to develop a direct and simple spectrophotometric method that could cope with the analysis of different types of environmental waters, within different salinity ranges (fresh to marine waters). The potential interference of metal ions, that can usually be present in environmental waters, was assessed and no significant interferences were observed (<10%). For a complete Cr(VI) determination (three replicas) cycle, the corresponding reagents consumption was 75 µg of DPC, 9 mg of ethanol and 54 mg of sulfuric acid. Each cycle takes about 5 min, including the system clean-up. The limit of detection was 6.9 and 12.2 µg L^−1^ for waters with low and high salt content, respectively. The method was applied for the quantification of chromium (VI) in both fresh and marine water, and the results were in agreement with the reference procedure.

## 1. Introduction

The determination of chemical species in different types of natural waters presents different challenges, depending on the water source. The most prevalent analytical challenges are the low analyte concentration and the complexity of the sample matrix. The sample characteristics can directly affect the accuracy of the analytical method, thus affecting the quality assurance of the data [1,2]. Water is a good example of how a sample can be addressed with different matrix composition (fresh, estuarine, marine or wastewater), having a direct impact in the analysis process.

Chromium can occur naturally in the environment in rocks, animals, plants, water and soil in different oxidation states. The most prevalent forms encountered in the environment are Cr(III) and Cr(VI). These two different chromium forms are an example on how the same element presents antagonistic physiological properties depending on the oxidation state. While Cr(III) is considered to be a micronutrient essential for living organisms, as it participates in metabolic processes, Cr(VI) is associated with a high toxicity, being carcinogenic [3,4,5,6]. In this scenario, the analytical speciation of chromium is more significant than the determination of the total content.

In general, the most used instrumental methods for the determination of metal ions are based on the determination of total content of the element: atomic absorption spectrometry (AAS), inductively coupled plasma-atomic emission spectrometry (ICP-AES) and, for ultra-low concentration, the electrothermal atomic absorption spectrometry (ETAAS) and inductively coupled plasma—mass spectrometry (ICP-MS) [7,8,9]. However, for chromium determination, these methodologies are less significant due to the importance of the speciation. A widely used method for hexavalent chromium determination is the conventional colorimetric method that resorts to the high selective colored reaction of Cr(VI) with diphenylcarbazide (DPC) [7].

A number of analytical methods have already been developed for chromium speciation in environmental waters, resorting to the spectrophotometric detection and automatic instrumental methods, such as flow-based methods [10,11,12,13,14,15,16,17,18,19,20,21], being one of those a flow injection analysis system published by the International Organization for Standardization (ISO 23913:2006). Flow-based methods are appealing to implement automatic and miniaturized methods with spectrophotometric detection due to the associated apparatus high versatility. The most associated features of a flow-based system are the use of low volumes of sample and reagents, reduced effluent production, minimization of sample/reagents handling, the increased sample throughput, and the increase of automation [22]. These features appear to be particularly important in the determination of contaminants.

In spite of the aforementioned advantages, these reported flow-based analytical papers [10,11,12,14,15,16,17,18,19,20,21] were only devoted to the analysis of low salt content environmental waters; only the reference material [13] mentions a possible application to seawater, if some adjustments are carried out. In this scenario, we propose a flow-based method for the quantification of Cr(VI) in different types of recreational waters (i.e., fresh to marine). The method was developed in a sophisticated and compact flow system from GlobalFIA, Inc. This all-in-one integrated equipment comprises all of the devices required for this type of systems including the propulsion system, the valve (with injection and selection valve mode), the flow cell, the light source, and the coupled charged device (CCD) detector. Additionally, it was designed to incorporate also the solutions (reagents and standards) needed for the analysis, the carrier and the waste containers, making it a system that can easily be adapted for the in-situ analysis, a point of high interest at the environmental analysis field. As a flow-based system, the configuration of the manifold can be easily adapted to different determinations. Furthermore, in the proposed method, the system was developed for the determination of chromium (VI) in both fresh and marine waters, with no need for laborious sample preparation (such as extraction procedures) before analysis.

## 2. Results and Discussion

The development of the flow-based system for the Cr(VI) determination involved several studies to assess the influence of some physical and chemical variables in the method performance. Those parameters were optimized in order to minimize the volumes of reagents and standards, increase the determination rate, and also attain the highest sensitivity for Cr(VI) determination (calibration curve slope). The studied parameters involved the volumes of standard and DPC solution, the concentration of DPC solution, the volume of ethanol for DPC dissolution, and the reaction coil length. 

### 2.1. Evaluation of the Flow Reaction Conditions

For the evaluation and optimization of the reaction conditions with standards prepared in MilliQ water (MQW), the preliminary studies were based on the conditions (reagent concentration and solutions volumes) set by Morais et al., (2002) [12]. The first study carried out was the influence of the DPC reagent and the standard/sample volumes on the method performance. The method performance was evaluated in terms of the sensitivity (slope of the calibration curve) and the calibration curve intercept. Different volumes of DPC reagent (25–75 µL) and standard/sample (350–600 µL) were tested and the calibration features evaluated (Figure 1). The increase on the standard/sample volume resulted in an increase on the calibration curve slope until 450.0 µL; no significant variation was observed for 500 µL (<10%). With higher volumes, the slope started to decrease. The selection between these two volume values were made based on the calibration curve intercept, the one with lower intercept was selected, that was achieved using 500.0 µL of standard/sample. Regarding the DPC reagent volume, the study started with the use of 75 µL of reagent. By using 75 or 50 µL of reagent no significant differences were observed in terms of the slope of the two calibration curves (<10%). However, the calibration curve intercept decreases about 40% when 50 µL was used. Comparing 50 and 25 µL, with the last, the slope decreases significantly (≈30%). The volumes set, represented with the solid-filled markers at the Figure 1, were 500 and 50 µL of standard/sample and reagent, respectively.

The PTFE reactor coil length was initially kept to a minimum (10 cm), allowing to physically connect the central port of the selection valve to the flow cell. However, a high relative standard deviation value between the different replicas of the same standard was observed, what can be due to the refractive index produced between the mixture of the two different matrix solutions, the DPC solution and the standard solution. As an attempt to lower the relative standard deviation between replicas, the reaction coil length was increased, and reaction coils of 45 and 60 cm were tested. With the increase of the reaction coil length, an increase of about 25% at the sensitivity was observed for both reaction coil lengths, but no significant differences were observed (<10%) when using 45 and 60 cm.

The DPC concentration and the DPC solution composition were also evaluated, with the main objective of minimize the reagents consumption. Concentrations of 0.025, 0.050 and 0.075% of DPC were tested. No significant differences in the calibration curves were observed when using the DPC with 0.050% or 0.075% in mass concentration. However, with the 0.025% concentration, the sensitivity of the determination decreased about 30%. The minimum volume of ethanol for the dissolution of the DPC salt was found to be 4% of the total reagent volume and the calibration curve was established with the different reagent dissolution matrix overlapped.

### 2.2. Interference Studies

The potential interference of some metal ions that can be present in water samples were tested (Table 1). The interference was assessed comparing the signal of a 100 µg L^−^^1^ Cr(VI) standard with the signal from a 100 µg L^−1^ Cr(VI) standard with the potential interfering species as interfering percentage (IP). For this study, the concentrations tested for all of the potentially interfering species was the maximum expected concentrations in streams and/or groundwater [7]. No significant interferences were observed for those values (Table 1), and so, higher concentrations of the potentially interfering species were tested. These higher concentrations did not also interfere with the Cr(VI) determination (IP < 8%). Regarding the presence of iron ions, and according to the ISO 23913:2006, it could interfere in the Cr(VI) determination for concentrations above 10 mg L^−1^, so several concentrations were tested. The presence of iron, in concentration up to 15 mg L^−1^, did not significantly interfere (IP < 6%).

The possible interference of the Cr(III) species on the Cr(VI) determination was also evaluated. Calibration curves with Cr(III) standards, prepared in the same concentration range of the Cr(VI) standards (25.0 to 200.0 µg L^−1^), were analyzed within the developed flow-based method. However, for all of the standards solutions, the signal has no statistical difference from the blank signal (<5%), and so the Cr(III) did not significantly interfere in the Cr(VI) determination.

One objective of this work was to develop a method for Cr(VI) determination that could cope with waters with different salinities, and so the influence of this parameter was studied. For that purpose, standards were prepared in MQW and in synthetic sea water.

When the standards were prepared in synthetic seawater, a refraction index gradient is produced at the interface between the different solutions (standards prepared in a high salt content matrix, and the DPC reagent prepared in ethanol and sulphuric acid). This affects the absorbance signal shape, when compared with the signal produced using standards prepared in MQW. The features of the calibration (slope and intercept) with the standards prepared in synthetic seawater are different from those obtained with standards prepared in MQW due to the aforementioned solution characteristics. When the standards were prepared in synthetic seawater, an increase of both values of the calibration features (slope and intercept) was observed. So, for the analysis of seawater samples, the standards should be prepared in artificial water, one of the recommendations of the ISO 23913:2006 [13].

### 2.3. Analytical Features

The analytical features of the developed flow-based system for the determination of Cr(VI) in fresh and sea water content are summarized in Table 2.

The limit of detection (LOD) and the limit of quantification (LOQ) values were calculated according to IUPAC recommendations as the concentration corresponding to the sum of three and ten times (for LOD and LOQ, respectively) the standard deviation to the mean value of ten consecutive blank solution measurements [23,24].

The repeatability was assessed by calculation of the relative standard deviation (RSD) of ten replicate analysis of a standard (100 µg L^−1^). The RSD was 1.5% for Cr(VI) determination for standards prepared in MQW and 3.0% for standards prepared in synthetic seawater.

A complete analytical cycle (three replicas) for the determination of Cr(VI) takes about 5 min. The corresponding reagents consumption for an analytical cycle was: 75 µg of DPC, 9 mg of ethanol, and 54 mg of sulfuric acid.

### 2.4. Application to Recreational Water—Method Validation

No chromium was detected in the different recreational water samples analysed by the developed flow-based system, the reference colorimetric method [7] and the ICP-AES (total chromium quantification). Consequently, these water samples were spiked with unknown quantities of chromium (VI) and the validation of the developed flow-based method was attained by the comparison of the obtained Cr(VI) concentration results with those obtained with the reference procedure [7].

The Cr (VI) concentrations obtained with the two procedures and the comparison results are displayed in Table 3. The relative deviation between the two set of results showed that there were no significant differences between the newly developed flow-based system and the reference procedure for the determination of Cr(VI) in waters with different salinities content (RD < 10%). Additionally, a regression equation was established between the two set of results, the Cr(VI) concentration obtained with the developed flow-based system ([Cr(VI)]_FA_ µg L^−1^) and the reference procedure ([Cr(VI)]_Ref_ µg L^−1^). The established regression equation was [Cr(VI)]_FA_ = 0.97 (±0.10) [Cr(VI)]_Ref_ + 2.15 (±10.80), where the values in brackets represents the 95% confidence interval. These parameters showed that the estimated slope an intercept did not differ statistically from 1 and 0, respectively [25].

### 2.5. Comparison with Some Previously Reported Spectrophotometric Flow-Based Methods

Some flow-based methodologies have been previously developed for the determination of hexavalent chromium in water samples using various chromogenic reagents (Table 4). The DPC reagent was the most used chromogenic reagent due to the associated selectivity for Cr(VI) species in acidic media [7]. However, unlike our proposed method, the previously published works were devoted for this quantification in waters with low salt content, and so, they have no applicability for seawater analysis. In fact, the ISO protocol [13] for the determination of Cr(VI) refers the possible application to seawater, but it requires some not specified adjustments to the method.

Regarding other methods without involving preconcentration processes, the low limits of detection achieved by Ma et al. [15] and Zhu et al. [10] could be due to the use of a liquid waveguide capillary flow cell with 500 and 250 cm optical path, respectively. In our work, the LOD is higher, but a simpler and potentially more robust flow cell was used. Some other works mention the use of an extraction procedure before the quantification, for the analyte preconcentration [11,18,19]. This procedure did not significantly improve the limits of detection and make the whole procedure more complex. Anyway, the limit of quantification of our method is far below the 1 mg L^−1^ proposed by the World Health Organization as screening value in recreational waters (WHO 2021) [26].

The herein proposed method was developed for the direct determination of hexavalent chromium, with no need for previous sample preparation, besides filtration. In comparison with the ISO procedure [13], it should be noted that the volumes of reagents and consequent effluents production were minimized by using a sequential injection strategy (this work), instead of the flow injection analysis assembly which implies a continuous reagents consumption. Additionally, the concentration of the reagents used (DPC and H_2_SO_4_) was lower and phosphoric acid was not used at all, as opposed to the ISO procedure and the colorimetric assay [7].

## 3. Material and Methods

### 3.1. Reagents and Solutions

All solutions were prepared with analytical grade chemicals and MilliQ water, MQW (resistivity > 18 MΩ cm, Millipore, Burlington, MA, USA).

A stock solution of 50.0 mg L^−1^ of Cr(VI) was prepared by dissolution of the corresponding quantity of potassium dichromate salt in water (K_2_Cr_7_O_7_, Merck, Darmstadt, Germany). An intermediate solution of 2.50 mg L^−^^1^ of Cr(VI) solution was prepared by dilution of the 50.0 mg L^−1^ stock solution. Working standards from 25.0 to 200.0 µg L^−1^ were weekly prepared by dilution of the 2.50 mg L^−1^ intermediate solution, with water and artificial seawater. 

Artificial seawater was prepared according to Kester et al. (1967) [27]. This seawater solution was composed by: 23.926 g kg^−1^ NaCl (Merck, Darmstadt, Germany), 4.008 g kg^−1^ Na_2_SO_4_ (Merck, Darmstadt, Germany), 0.677 g kg^−1^ KCl (Merck, Darmstadt, Germany), 0.196 g kg^−1^ NaHCO_3_ (Merck, Darmstadt, Germany), 0.098 g kg^−1^ KBr (Merck, Darmstadt, Germany), 0.026 g kg^−1^ H_3_BO_3_ (Aldrich, St Louis, MO, USA), 0.003 g kg^−1^ NaF (Merck, Darmstadt, Germany), 0.05327 mol kg^−1^ MgCl_2_·6H_2_O (Merck, Darmstadt, Germany), 0.01033 mol kg^−1^ CaCl_2_·2H_2_O (Merck, Darmstadt, Germany), and 0.00009 mol kg^−1^ SrCl_2_·6H_2_O (Sigma-Aldrich, St. Louis, MO, USA).

A 0.75 mol L^−1^ sulfuric acid solution was prepared by dilution of the concentrated solution (d = 1.804, 95.0–97.0%, Fluka, Germany).

A 0.050% of diphenylcarbazide (DPC) solution was prepared by dissolution of the corresponding quantity of the reagent (1,5-diphenylcarbazide, Sigma-Aldrich, Darmstadt, Germany) in 3.0 mL of ethanol, 2.0 mL of water and made up to 50.0 mL with sulfuric acid 0.75 mol L^−1^.

All solutions used for the interferences assessment (Al(III), Ca(II), Cd(II), Co(II), Cr(III), Cu(II), Fe(III), Mg(II), Mn(II), Ni(II), Pb(II) and Zn(II)) were prepared by diluting the commercial atomic absorption standard solution (1000 mg L^−1^, Merck, Darmstadt, Germany).

### 3.2. Apparatus

A FloPro Researcher equipment controlled by FloZF data acquisition and device control software (GlobalFIA, Fox Island, WA, USA) was used. The fluidics manifold was equipped with a bi-directional milliGAT™ pump, connected to the central channel of a 10-port multi-position selection valve (VICI Cheminert^®^ C25Z-3480-M17, Houston, TX, USA) and an absorbance flow cell with 50 mm optical path. As detection system, an Ocean Optics USB 4000 charged coupled device (CCD) spectrophotometer detector was coupled to the absorbance flow cell and the polychromatic light source using fiber optic cables. All of the components of the flow system were connected by PTFE tubing from Omnifit^®^ (0.8 mm i.d., Merck, Darmstadt, Germany).

### 3.3. Flow Manifold and Procedure

The developed flow manifold for the spectrophotometric determination of Cr(VI) in waters is depicted in Figure 2.

The sequence of steps for Cr(VI) determination and respective volumes are shown in Table 5. The reagent and the sample were aspirated to the holding coil (steps A and B), and subsequently the plugs were sent to the detector (step C) for Cr(VI) detection. Each sample determination, corresponding to a cycle, comprises three replicas. At the end of each replica, the flow system was washed with sulfuric acid 0.75 mol L^−1^ (steps D and E).

### 3.4. Water Sample Collection and Preparation

Water samples from various recreational locations from Porto district (Portugal) were collected 20 cm from the surface. The samples were filtered with Acrodisc 25 mm syringe filters 0.45 µm (Pall, New York, NY, USA) and analysed within the first 24 h after collection, according to the reference sampling procedure [7]. Samples were kept refrigerated until analysis.

Superficial, bottom and interstitial water samples from Douro river estuary were collected in 500 mL acid-cleaned polyethylene bottles, along the salinity gradient at rising high tide.

Key physical and chemical parameters, namely temperature, conductivity, salinity, pH, oxygen saturation and turbidity were measured in situ using a YSI6920 CTD multiparameter probe.

### 3.5. Reference Procedure

For comparison purposes, the determination of chromium (VI) was carried out in parallel by the colorimetric method according to the reference method described in the “Standard Methods for the Examination of Waters and Wastewater” [7], for hexavalent chromium determination. Results were compared with those obtained with the developed flow-based method.

## 4. Conclusions

The main feature of the method herein proposed is the possibility of the determination of Cr(VI), in a single manifold, in different types of water, including seawaters. Furthermore, the use of an integrated and compact flow-based system, the GlobalFIA equipment, was also a point of high interest of the developed analytical method for environmental monitoring. As the system displays all of the required flow components (propulsion system, selection and injection valve, flow cell, light source, and the detector) in a small, robust and portable equipment (20 cm wide by 20 in depth and 15 in height), it can be used for the in situ environmental analysis.

## Figures and Tables

**Figure 1 molecules-27-02073-f001:**
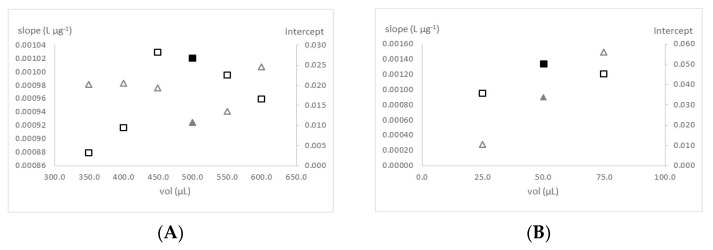
Study of the influence of the sample volume (**A**) and DPC reagent volume (**B**) on the method sensitivity, expressed as the calibration curve slope (squares), and on the calibration curve intercept (triangles). The chosen values are represented by solid-filled markers.

**Figure 2 molecules-27-02073-f002:**
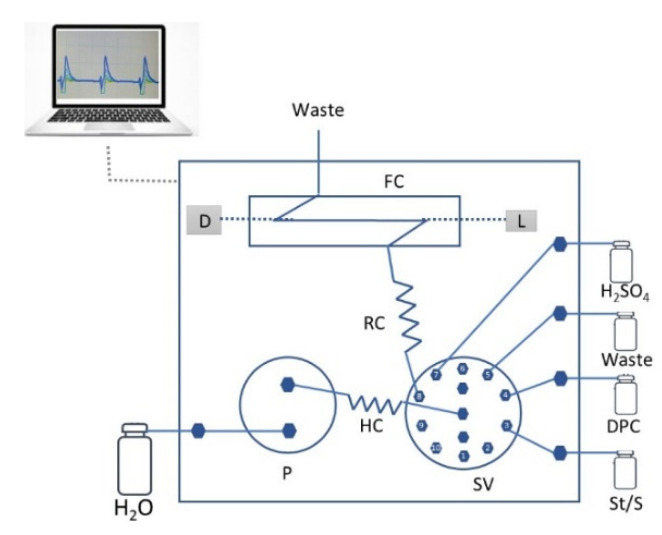
Flow manifold for Cr(VI) determination in water. St/S—standard or sample solution; R1, DPC solution (0.050%); P, syringe pump; SV, selection valve; HC, holding coil (300 cm); RC, reaction coil (50 cm); D, Ocean Optics USB 4000 CCD detector; L, light source; FC, flow cell (50 mm path length); W, waste.

**Table 1 molecules-27-02073-t001:** Study of the interference of some metal ions, commonly present in natural waters, in the Cr(VI) determination. Comparison of the absorbance signal between a 100 µg L^−1^ Cr(VI) standard with the signal from a 100 µg L^−1^ Cr(VI) standard with the potential interfering species as interfering percentage.

Tested Ion	Conc. in Water * µg L^−1^	Tested Conc. µg L^−1^	Interference in Cr(VI) Determination%	Tested Ion	Conc. in Water * µg L^−1^	Tested Conc. µg L^−1^	Interference in Cr(VI) Determination%
Al^3+^	400	400	7.6	Mg^2+^	4000	4000	3.7
		800	−1.5			8000	−6.6
		1600	−1.3			16,000	0.1
Ca^2+^	15,000	15,000	1.7			50,000	−5.7
		30,000	−1.5	Mn^2+^	100	100	2.6
		60,000	0.7			200	2.8
Cd^2+^	10	10	2.2			400	1.2
		20	5.5	Ni^2+^	100	100	4.7
		40	5.9			200	5.7
Co^2+^	10	10	4.9			400	5.3
		20	5.3	Pb^2+^	100	100	1.7
		40	1.0			200	0.8
Cu^2+^	100	100	2.3			400	1.9
		200	2.6	Zn^2+^	100	100	2.0
		400	2.3			200	5.1
Fe^3+^	700	700	5.2			400	2.5
		1400	1.7				
		3000	3.2				
		10,000	2.1				
		15,000	2.5				

* Highest expected ions concentrations in streams and/or groundwaters [7].

**Table 2 molecules-27-02073-t002:** Features of the developed flow-based system for Cr(VI) quantification in fresh and seawater; dynamic range, 25 to 200 µg L^−1^; LOD, limit of detection; LOQ, limit of quantification; SD, standard deviation.

Sample Type	Typical Calibration Curve ^a^ A = (Slope ± SD) µg L^−1^ Cr(VI) + Intercept ± SD	LOD (µg L^−1^)	LOQ (µg L^−1^)
Freshwater ^b^	A = (1.23 × 10^−3^ ± 1 × 10^−5^) [Cr(VI)] + 0.007 ± 0.002	6.9	10.8
Seawater ^c^	A = (1.48 × 10^−3^ ± 7 × 10^−5^) [Cr(VI)] + 0.015 ± 0.009	12.2	27.5

^a^ n = 3 calibration curves from different days. ^b^ standards prepared in MQW. ^c^ standards prepared in artificial seawater.

**Table 3 molecules-27-02073-t003:** Chemical parameters of the analyzed water samples and the results obtained with the developed flow-based system for Cr(VI) determination ([Cr(VI)]_FA_) and with the reference procedure ([Cr(VI)]_Ref_); G, conductivity; SD, standard deviation; RD, relative deviation.

Sample ID	pH	G (µS/cm)	Salinity		[Cr(VI)]_FA_ ± SD µg L^−1^	[Cr(VI)]_Ref_ ± SD µg L^−1^	RD%
F1	7.33	1054	0.53	estuarine	54.4 ± 3.7	51.2 ± 1.9	6.2
F2	7.44	1109	0.56	estuarine	34.4 ± 0.8	37.6 ± 1.4	−8.5
F3	6.92	363	0.19	freshwater	38.0 ± 2.0	37.6 ± 1.4	1.1
F4	7.84	550	0.36	estuarine	51.8 ± 1.5	50.8 ± 1.9	2.0
F5	7.63	2013	1.22	estuarine	84.0 ± 7.6	92.1 ± 3.4	−8.8
F6	7.58	2542	1.55	estuarine	109 ± 1	103 ± 4	5.3
F7	7.48	283	0.16	estuarine/fresh	102 ± 2	94.9 ± 3.5	6.6
M1	7.72	50,252	33.94	coastal marine	95.4 ± 2.0	94.8 ± 3.5	0.6
M2	7.87	51,322	33.68	coastal marine	134 ± 3	129 ± 5	3.9
M3	7.73	52,809	34.76	coastal marine	126 ± 2	122 ± 5	3.5
M4	7.87	50,629	33.21	coastal marine	119 ± 3	130 ± 5	−8.6
M5	7.72	50,252	33.94	coastal marine	130 ± 7	130 ± 5	−6.1
M6	7.87	51,322	33.68	coastal marine	147 ± 2	141 ± 5	−1.4
M7	7.87	50,629	33.21	coastal marine	156 ± 5	162 ± 6	−3.6

**Table 4 molecules-27-02073-t004:** Analytical characteristics of developed spectrophotometric flow-based systems for chromium (VI) determination in water samples (presented in descending chronological order).

System	Type of Water	Sample (µL)	Reagent	Sample Throughput (h^−1^)	LOD (µg L^−1^)	Refs.
FA	Fresh Marine	500	DPC	36	6.9 12.2	This work
iSEA	Tap, river, industrial waste and bottled	2100	DPC	30	1.25/0.028 *	[10]
FIA	Waste, fresh and river	800	DPC	12	1.25	[11]
SIA	Ground and waste	800	DPC	_	10	[14]
FIA	Drinking	350	DPC	30	0.0078	[15]
SIA	Tap and well	100	H_2_O_2_	_	600	[16]
rFIA	Waste	_	MB, SP		7	[17]
MCFIA	River and spring	96	DPC	105	1.0	[18]
SIA	Simulated water	300	DPC	53	2.4	[19]
FIA	Tap and mineral	100	CA	_	1.0	[20]
CFA	Surface waters	_	DPC	8	48	[21]
SIA	Waste	2555	DPC	40	30	[12]

* Different limits of detection correspond to the use of different flow cells. FA, flow-based system; DPC, 1,5-diphenylcarbazide; iSEA, integrated syringe pump-based environmental water analyzer; FIA, flow injection analysis; SIA, sequential injection analysis; rFIA, reverse flow injection analysis; MB, methylene blue; SP, sodium periodate; MCFIA, multicommutated flow analysis; CA, chromotropic acid; CFA, continuous flow analysis.

**Table 5 molecules-27-02073-t005:** Protocol sequence for the Cr(VI) determination in recreational waters.

Step	SV Position	Volume (µL)	Flow-Rate (µL s^−1^)	Description
A	4	50	30	Aspirate DPC 0.050% reagent solution
B	3	500	30	Aspirate standard/sample solution
C	8	1500	30	Propel through the flow cell for Cr(VI) quantification
Clean-up step				
D	7	200	40	Aspirate H_2_SO_4_ solution
E	8	700	40	Propel through the flow cell

## Data Availability

Not applicable.

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
