# Peer review of "A Robust Flow-Based System for the Spectrophotometric Determination of Cr(VI) in Recreational Waters"

_molecules, 2022, doi:10.3390/molecules27072073_

Round 1
Reviewer 1 Report
The present manuscript reports on the development of flow-based method for the spectrophotometric determination of chromium (VI) in waters with different salinities. The article is quite well-written and contains interesting results. However, some imprecisions are present, and the manuscript needs major revisions to be accepted for publication. See details here below.
- (Lines 50-56) ICP-MS based methods for chromium speciation in water samples must be mentioned in the Introduction. In particular, HPLC-ICP-MS methods (DOI: 10.3390/molecules24040668, 10.1016/j.talanta.2015.09.047, 10.1016/j.aca.2016.03.039) and a recently developed frontal chromatography-ICP-MS method (DOI: 10.1016/j.jhazmat.2021.125280) for the selective determination of Cr(VI) in water must be cited to provide a complete state-of-the-art on this topic. Please add these, and possibly also other additional, relevant references in the manuscript.
- (Table 1) The interference expressed as relative % is meaningless in the absence of the Cr(VI) concentration value (e.g., an error of 7.6% in determining a Cr(VI) concentration of 100 or 10’000 μg/L is markedly different!). Please provide this value together with the absolute error in quantification expressed in μg/L.
- (Line 161-172) This part of the text is not clear. Synthetic seawater is stated to be unsuitable for the sake of calibration owing to signal interferences. It is expected that the same interferences are present also for natural seawater samples owing to the similar composition. It must be proven that the calibration features obtained by using artificial water (ISO 23913:2006) are the same achieved by using a natural seawater sample to support the choice of the artificial water matrix for calibration. Alternatively, the method of standard addition may be proposed for complex water samples such seawater.
- The subsection 2.3 should be renamed “Analytical features”.
- (Table 3) The unit of measurement of “Salinity” is missing.
- (Table 3) The last column should be expressed as “Recovery (%)” (100% means [Cr(VI)]REF = [Cr(VI)]FA) as commonly done in the literature.
- The conclusions section mostly contains discussion regarding the comparison with existing methods. Only the text from Line 291 to 298 can be considered as conclusions. The rest of the text (Lines 299-332) should be moved to a new section in the Results and Discussions (section 2.5 “comparison with existing methods”).
- (Table 5) The comparison of the analytical features achieved with ICP-MS based methods (see works mentioned in the comment 1) and with the proposed method should be added in this Table or in the discussion.
Reviewer 2 Report
This is a well-structured paper reporting in that authors propose a method for the quantitative spectrophotometric determination of Cr(VI) in natural recreational waters with different salinities - fresh to marine water. This method is developed in a compact flow system from GlobalFIA Inc, USA. The system is adjusted according to the proposed method. The method is automated and is applied by use of integrated all-in-one portable equipment. The main advantage of this device is the fast and efficient Cr(VI) determination with minimal reagents consumption, with no need for laborious sample preparation. It is adapted for the in situ environmental analysis.
Recommendations:
- The linear relationship - item 2.4, line 209, is not clear - when validating the method, the authors should present the regression equation.
- In line 210 in the record: [Cr(VI)]FA= 0.97 ± 0.10 [Cr(VI)]Ref +2.15 ± 10.80, there is a technical error (bold).
- The data in Table 3 for columns: [Cr(VI)]FA and [Cr(VI)]Ref should be presented as: X average ± SD, so an interval estimate can be made.
Round 2
Reviewer 1 Report
All requested changes have been made, therefore the article can be accepted for publication.